**Subject Category:**
Biology (whole organism)

behaviour/ecology/evolution

noise pollution, host–parasitoid interactions, *Ormia ochracea*, soundscape, *Gryllus lineaticeps*, experimental playback

**Authors for correspondence:**
Jennifer N. Phillips
e-mail: jnphilli@calpoly.edu
Clinton D. Francis
e-mail: cdfranci@calpoly.edu

†These authors contributed equally to this study.

# Background noise disrupts host–parasitoid interactions

Jennifer N. Phillips†, Sophia K. Ruef†, Christopher M. Garvin†, My-Lan T. Le and Clinton D. Francis

Department of Biological Sciences, California Polytechnic State University, San Luis Obispo, CA 93407, USA

JNP, 0000-0002-0014-2995; CDF, 0000-0003-2018-4954

The soundscape serves as a backdrop for acoustic signals dispatched within and among species, spanning mate attraction to parasite host detection. Elevated background sound levels from human-made and natural sources may interfere with the reception of acoustic signals and alter species interactions and whole ecological communities. We investigated whether background noise influences the ability of the obligate parasitoid *Ormia ochracea* to locate its host, the variable field cricket (*Gryllus lineaticeps*). As *O. ochracea* use auditory cues to locate their hosts, we hypothesized that higher background noise levels would mask or distract flies from cricket calls and result in a decreased ability to detect and navigate to hosts. We used a field manipulation where fly traps baited with playback of male cricket advertisement calls were exposed to a gradient of experimental traffic and ocean surf noise. We found that increases in noise amplitude caused a significant decline in *O. ochracea* caught, suggesting that background noise can influence parasitoid–host interactions and potentially benefit hosts. As human-caused sensory pollution increases globally, soundscapes may influence the evolution of tightly co-evolved host–parasitoid relationships. Future work should investigate whether female cricket phonotaxis towards males is similarly affected by noise levels.

## 1. Introduction

Acoustic signals and cues are critical for interactions within and among species, and the acoustic backdrop within which they propagate has the potential to influence the outcome of myriad interactions. This is especially true in the context of evolutionarily novel acoustic conditions created by anthropogenic noise [1], and insects have been vastly understudied in this context [2]. Although natural background sounds from moving water, wind and biotic choruses are known to be selective agents shaping acoustic signalling strategies, rapidly expanding human population is leading to unprecedented soundscapes within

**Figure 1.** Power spectra of traffic (red) and ocean surf (blue) illustrate strong masking potential of cricket advertisement calls (teal). Traffic and ocean surf noise files were standardized to the same peak amplitude prior to comparison.

populated regions as well as heavily protected natural areas [3]. Considerable research over the last two decades has revealed that human-made noise can impair intraspecific communication used for mate attraction [4] or threat detection [5], alter animal vigilance and foraging [6,7], and fundamentally restructure communities [8,9]. A limited number of studies have explored the influence of anthropogenic noise on acoustically oriented predators (e.g. [10–12]), and fewer have focused on other interspecific interactions, such as mutualism or parasitism [8,13,14]. Anthropogenic noise may be especially influential for predator–prey and host–parasite/parasitoid interactions that involve eavesdropping, where unintended listeners intercept cues or signals propagating from hosts, prey or predators. Bats and frog-biting midges (*Corethrella* spp.) are well known to eavesdrop on anuran mating signals when localizing prey [15,16], predators use nestling begging calls to localize nests [17], and mate attraction signals may be used by avian brood parasites, such as cowbirds, to locate and assess potential host nests [18]. How these interactions are influenced by changes to the acoustic environment due to human activities is not well understood (but see [14]), especially among host–parasite or parasitoid interactions.

One well-studied host–parasitoid interaction is between the variable field cricket (*Gryllus lineaticeps*) and the tachinid fly *Ormia ochracea*. Female flies eavesdrop on male variable field cricket mating calls and lay their larvae on and near the male cricket, which burrow into the host and develop over a 7-day period before emerging and killing the host in the process [19]. Female crickets prefer faster and longer calls, as do the parasitoids, thereby creating a sexual conflict of attracting females at the cost of being parasitized [20]. These types of host–parasitoid interactions are classic examples of an evolutionary arms race, where both host and parasitoid typically co-evolve antagonistically [21,22]. For example, in an island population of field crickets (*Teleogryllus oceanicus*), males became silent over 20 generations probably due to selection pressure after an introduction of *O. ochracea* [23]. While this type of rapid evolution has not been observed in mainland species, there are often smaller changes in environments or parasitoid abundance that change behaviours and probably the success or demise of parasitoids and hosts (e.g. [24]). In captive experiments, male field crickets do not seem to slow their call rate when parasitoids are present [25]; rather, males from high-risk populations tend to increase their call rate, perhaps to 'hurry up' and mate before they die of parasitism later in the season [26]. However, female crickets can also be parasitized, which may lower their choosiness for faster calls [27]. A great deal of research has focused on how these flies hear and orient toward the cricket song. Given their small size, they are able to localize a host with two degree accuracy by amplifying sound differences, similar to the hearing system of humans [28,29]. However, it is unknown whether background noise interferes with the ability of these flies to detect cricket songs, or if crickets could avoid parasitism by living in noisier areas.

Background noise from both natural and anthropogenic sources consists of a wide range of frequencies that overlap with animal vocalizations, which can mask those signals from being detected by receivers [30]. Typically, ocean noise and traffic noise are louder at low frequencies with decreasing energy at higher frequencies (figure 1). Therefore, most studies focus on how low-frequency noise overlaps low-frequency signals. For example, the call of the túngara frog is relatively low (1–3 kHz),

and frog-biting midges (*Corethrella* sp.) that orient toward túngara calls are less abundant in noise [14]. Similarly, the relatively high acoustic energy in traffic and ocean surf noise at 5 kHz overlaps the peak frequency of cricket calls (figure 1). Given this overlap, these sources of background noise are likely to at least partially mask the signal for receivers (figure 1).

Here, we used a manipulative field experiment to determine how altered acoustic regimes influence variable field cricket host localization success in the tachinid parasitoid fly (*O. ochracea*). Given that *O. ochracea* uses phonotaxis to localize suitable hosts, and because anthropogenic noise represents a strong masking potential to cricket calls (figure 1), we predicted that increases in traffic noise levels negatively affect *O. ochracea* phonotaxis, resulting in fewer flies caught at cricket playback speakers exposed to higher noise levels. Finally, because many environments experience high levels of noise from natural sources, we also carried out trials using ocean surf noise and expected a similar relationship between noise levels and number of flies caught. If noise decreases parasitism, variable field crickets may actually benefit from vocalization near noisy areas, at least in terms of reduced parasitism.

# 2. Material and methods

## 2.1. Study area

Our study took place from August to September in 2016 and 2017 in the Santa Monica Mountains of California, which is the breeding season for variable field crickets [31]. The Santa Monica Mountains are a transverse range in Southern California and consist of chaparral and oak woodlands. This area has two predominant ecosystems: coastal sage scrub along the coast, which transitions to California montane chaparral and woodlands as the mountains rise and recede from the coast and descend into the interior valleys. The highest abundance of *O. ochracea* was counted in the montane chaparral ecosystem, as determined by pilot trials; therefore, we conducted all playbacks in chaparral scrublands.

## 2.2. Cricket advertisement call and noise playback preparation

We recorded calls from 23 *G. lineaticeps* males on California Polytechnic State University-owned lands adjacent to San Luis Obispo, California using a Marantz PMD660 digital recorder (sample rate = 44.1 kHz; bit rate = 16); and an Audio Technica AT 815 microphone. Calls were recorded for at least 30 s at a distance of approximately 1 m. We then edited each file in Audacity 2.0.6 to remove background noise below 3000 Hz, eliminate other non-cricket sounds (i.e. birdsong) and standardized the peak amplitude across all files. We retained unique advertisement call variation from the different recorded males, such that the chirp rate was variable but representative of the sampled variance (mean ± s.d. chirps per second = 2.49 ± 0.72, range = 1.33–3.67). Therefore, the 20 randomly selected files used in our trials (see below) we varied somewhat in total length (mean ± s.d. seconds = 28.82 ± 8.46, range = 18.5–52.66), and fall within the normal range of cricket song frequency (3–8 kHz).

To simulate anthropogenic noise, we recorded six traffic noise samples throughout San Luis Obispo County using Roland R05 recorders at a distance of 10 m from roadways. Recordings averaged 200 ± 16 (s.e.) seconds and were standardized to the same peak amplitude in Raven Pro 1.5 [32]. To each recording, we added a 5 s fade in and fade out to the beginning and end of each to eliminate rapid onset and offset as the recording playbacks looped continuously. To explore the possible difference between traffic noise and natural sources of background noise, we used ocean surf noise in a subset of experimental trials. To simulate ocean surf noise, we recorded ocean surf noise at four locations throughout the Central Coast of California at 10 m from the edge of coastal bluffs using identical methods to those describe for traffic noise. To provide a stimulus typical of coastal-scrub environments on the Central Coast of California, we combined the four recordings into a single 60 min playback file and cross-faded the original four recordings with 5 s fade in and fade out so that the transition from one recording to the next was seamless.

## 2.3. Noise gradient and traps

Because *O. ochracea* activity is highest shortly after dusk [25], trials began approximately 20 min after sunset. We used ION Block Rocker (frequency response: 65 Hz–20 kHz) speakers in 2016 and TIC GS5 speakers (frequency response: 55 Hz–20 kHz) in 2017. Different speakers were used in 2017 because of internal battery failure in the ION Block Rocker speakers between seasons. During each trial, the

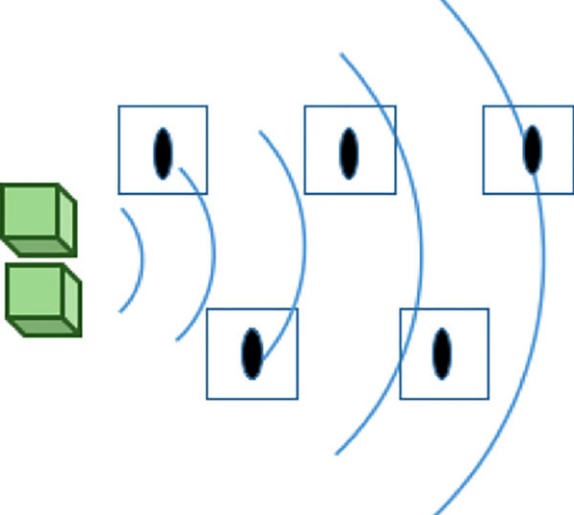

**Figure 2.** Schematic of general experimental design. Green cubes denote the speaker (either 2 TIC GS5 or one Block Rocker speaker). Each black oval represents a speaker broadcasting a unique cricket advertisement call and blue squares represent wood panels treated with Tanglefoot.

speakers played a unique traffic recording to establish a gradient in background noise levels, which ranged from 39.2 to 65.8 dB(A) $L_{eq}$. Noise playback speakers were set within 75–80 dB(A) at 1 m measured using a Larson-Davis 831 sound level meter (3 min, equivalent continuous sound level $L_{eq}$, *fast response*, re: 20 µPa). For each trial, 5–10 wood tiles (23 × 28 cm) prepared with a thin layer of a sap-based adhesive (i.e. Tanglefoot, Deerfield, MI, USA) [16] were set out such that all traps were typically 8 m apart (at least 5 m; figure 2). This spacing ensured that traps spanned over two orders of magnitude in background noise levels (approx. 40–65 dB(A)), which is within the range of noise levels known to influence the behaviour and ecology of others species [33]. Ambient noise levels were measured at each trap as described above prior to turning on any cricket playbacks. From the centre of each tile, we used Satechi SD Mini (frequency response: 90 Hz–20 kHz; figure 2) speakers to play back a randomly selected cricket recording standardized to a peak amplitude of 81 dB(A) at 10 cm, which corresponds to natural call amplitude [27]. Each trial duration lasted for 2 h at which point we counted the number of *O. ochracea* trapped on each tile.

Because we used two systems for background noise reproduction, we verified that there were no large differences in the reproduction of these stimuli between systems. First, we compared TIC GS5 and Block Rocker background noise reproduction by playing a randomly selected traffic file through both systems set to the same peak amplitude and recorded at a uniform distance of 3 m using a Roland R05 recorder with a micW i436 omnidirectional microphone. Power spectra comparisons of noise files standardized to identical peak amplitudes suggest that the two speaker systems produced background noise with similar masking potential for cricket advertisement calls and performed fairly well at reproducing frequencies of the original traffic recording (figure 3).

## 2.4. Data analysis

To determine whether ambient noise levels impact the ability of *O. ochracea* to detect cricket calls, we used linear mixed effect models (LMMs) in the *lme4* package in R [34]. In preliminary analyses, we had explored generalized LMMs with Poisson error, given that number of flies caught reflects counts. However, models consistently suffered from overdispersion. Therefore, we used LMMs and for each model we log transformed number of *O. ochracea* following a quantitative adjustment of adding one to all observations. We first constructed a null model that incorporated trial to account for the potential influence of environmental conditions on a particular sampling night and cricket playback stimulus to account for variation in cricket call characteristics (see below) as random effects. We then used likelihood-ratio tests to determine whether the addition of noise level improved model goodness of fit. Additionally, we explored whether inclusion of trials where traps were exposed to a gradient of ocean surf noise, which are spectrally similar to traffic noise (figure 1), resulted in the same pattern. We tested whether including a parameter that identified the type of background noise (i.e. traffic versus ocean surf) improved model

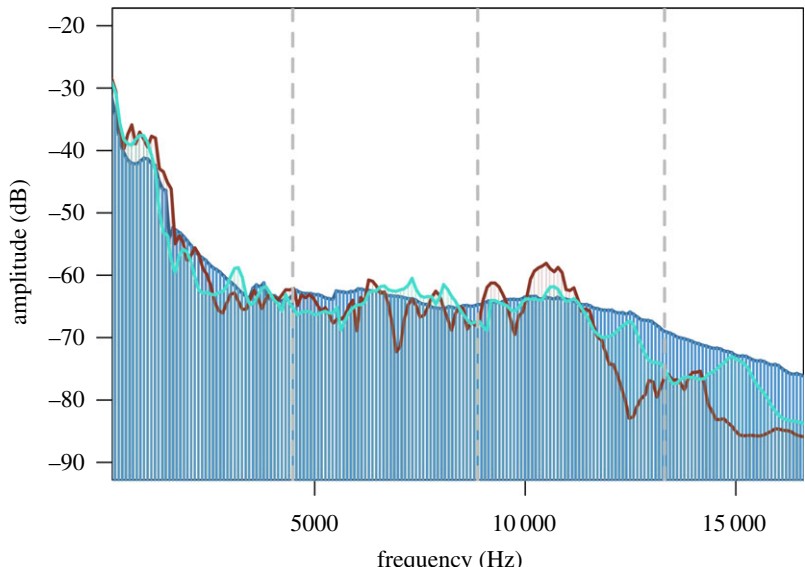

**Figure 3.** Power spectra comparison of a single original traffic file recording (blue) and the same recorded following playback from TIC GS5 speakers (teal) and Block Rocker (red). Vertical dashed lines correspond to signal peaks for cricket calls.

fit using a likelihood-ratio test. Additionally, because we used two different speakers between the 2 years to broadcast noise, we tested whether including speaker type as a fixed effect improved model fit. Finally, because male cricket advertisement call rate is known to influence parasitoid risk [20,35], we also explored whether variation in call rate among our cricket call playback stimuli influenced model performance in two ways with the trials with only traffic noise as the background stimulus and for trials that had traffic noise and ocean surf as background stimuli. First, to models that included noise level as a fixed effect and trial and cricket playback stimuli as random effects, we added the fixed effect of call rate to determine whether the inclusion of this parameter improved model fit. Next, we considered the interaction between call rate and noise. Lastly, because the random effect of cricket playback stimuli may capture variation in call rate, as well as other call characteristics, we removed cricket playback stimuli as a random effect and repeated the addition of call rate to models with noise level as a fixed effect.

# 3. Results

## 3.1. Noise level affects the number of flies caught

In total, we set 60 tile traps across eight traffic noise trials. We found that inclusion of noise level at each trap improved model fit relative to the model that only included random effects ($\chi^2 = 5.767$, $p = 0.016$, d.f. = 1) and that increases in traffic noise negatively influenced number of *O. ochracea* caught (table 1). Analyses including traps in a gradient of traffic noise ($n = 60$) or ocean surf noise ($n = 18$) resulted in the same pattern; increases in noise level negatively influenced number of *O. ochracea* caught ($n = 78$; $\chi^2 = 7.857$, $p = 0.005$, d.f. = 1; figure 4 and table 2) such that the estimated number of flies caught at the loudest sound level (approx. 65 dB) was more than 50% lower than the estimated number caught in the most quiet condition.

## 3.2. Both natural and anthropogenic noise decreases the number of flies caught

The inclusion of background noise type (i.e. ocean surf versus traffic) as an additional predictor did not further improve model fit ($\chi^2 = 0.020$, $p = 0.887$, d.f. = 1), suggesting that both sources have similar consequences for host localization and phonotaxis in female *O. ochracea*.

## 3.3. Speaker type, call rate and the interaction between call rate and noise does not influence fly abundance

Additionally, inclusion of speaker type as a fixed effect did not significantly improve model fit for traffic only ($\chi^2 = 3.183$, $p = 0.074$, d.f. = 1) or for traffic and ocean noise trials combined ($\chi^2 = 2.684$, $p = 0.101$,

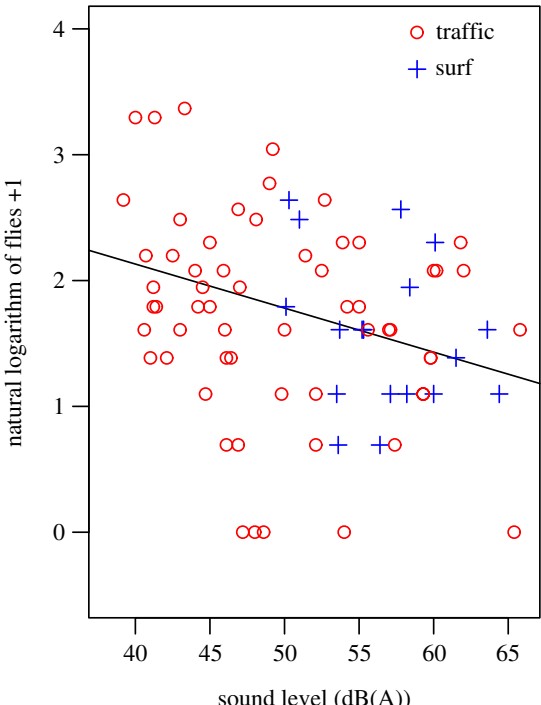

**Figure 4.** Increases in traffic noise or ocean surf sound resulted in fewer *O. ochracea* caught at cricket advertisement call traps. *Y*-axis reflects the natural logarithm of number of flies caught plus one.

**Table 1.** Final model parameters for traffic noise trials.

| random effects | variance | s.d. | | | |
|---|---|---|---|---|---|
| stimulus | 0.069 | 0.263 | | | |
| trial | 0.209 | 0.457 | | | |
| residual | 0.357 | 0.597 | | | |
| fixed effects | estimate | s.e. | d.f. | *t*-value | *p* |
| (intercept) | 3.496 | 0.706 | 59.970 | 4.954 | <0.001 |
| noise level (dB(A)) | −0.034 | 0.014 | 58.080 | −2.517 | 0.014 |

**Table 2.** Final model parameters for both ocean and traffic noise trials.

| random effects | variance | s.d. | | | |
|---|---|---|---|---|---|
| stimulus | 0.066 | 0.258 | | | |
| trial | 0.173 | 0.416 | | | |
| residual | 0.328 | 0.572 | | | |
| fixed effects | estimate | s.e. | d.f. | *t*-value | *p* |
| (intercept) | 3.531 | 0.629 | 75.787 | 5.615 | <0.001 |
| noise level (dB(A)) | −0.035 | 0.012 | 76.934 | −2.940 | 0.004 |

d.f. = 1). Finally, inclusion of cricket playback call rate as a fixed effect did not improve model performance (traffic only, $\chi^2 = 0.486$, $p = 0.485$, d.f. = 1; traffic and ocean noise, $\chi^2 = 0.630$, $p = 0.427$, d.f. = 1), nor did the interaction between call rate and noise improve model fit relative to a model with noise level only (traffic only, $\chi^2 = 1.262$, $p = 0.532$, d.f. = 2; traffic and ocean noise, $\chi^2 = 1.174$, $p = 0.556$, d.f. = 2). Similarly, call rate without playback stimuli as a random effect did not improve model fit over the model with just noise as a fixed effect (traffic only, $\chi^2 = 0.918$, $p = 0.338$, d.f. = 1; traffic and ocean noise, $\chi^2 = 1.061$, $p = 0.303$, d.f. = 1).

# 4. Discussion

Our results add to a growing body of literature that emphasizes the importance of the acoustic environment to wild organisms and species interactions (reviewed in [1,33]). Just as female frog navigation and orientation towards calling males is impaired by traffic noise [36], and many acoustic predators, such as owls and bats, have difficulty hunting prey in noisy environments [11,12,37], our results suggest that elevated background noise levels impair phonotaxis of specialized acoustic parasitoids when seeking hosts, potentially indicating a benefit to cricket hosts in noisy areas. Additionally, our findings contribute to a relatively small body of the literature documenting responses to altered noise regimes among insects (e.g. [38]).

Variable field cricket (*G. lineaticeps*) calls generally range from 3 to 8 kHz and *O. ochracea* ears detect and localize frequencies at their highest neural sensitivities between 4 and 9 kHz [39]. Although most traffic and ocean noises are low frequency, substantial energy extends into *O. ochracea*'s best sensitivity and the peak frequency of variable field cricket calls (figure 1), which could decrease signal-to-noise ratios and render signal detection and localization difficult for the parasitoids via masking [30]. Low-frequency noise may also act as a distraction from the target cricket calls [40,41], as evidenced by crabs that change their risk assessment behaviour in noise to predators [42]. Additionally, even if low-frequency noise does not directly mask cricket calls, it may impair the parasitoid's ability to detect variation in temporal patterns of the calls, which is demonstrated to be an important feature for *O. ochracea* attraction [20,35,43].

Our results suggest that hosts could benefit from less parasitism when living in a noisy area. To the best of our knowledge, only two studies suggest that altered noise regimes may influence host–parasite interactions. For example, near Gamboa, Panama frog-biting midges were less abundant in real and simulated noise-polluted areas [14]. Similarly, in a landscape exposed to energy sector noise in New Mexico, USA, nest parasitism by brown-headed cowbirds (*Molothrus ater*) was largely restricted to quiet locations [8]. However, it is unclear whether patterns of cowbird nest parasitism were impacted by noise *per se* or if parasitism patterns reflected distributions of the most-favoured hosts, which also tended to avoid noise. Additional work should investigate whether crickets are more abundant in noise and have fewer parasitoid infections to support the hypothesis that hosts may benefit from noise.

Our finding that risk of parasitoid infection may depend on acoustic conditions adds to a small body of work in animal behaviour and evolutionary biology. Relaxed selection from parasitoids could produce heritable changes to cricket traits by allowing other selective agents to operate [44], such as male advertisement calls that match female preferences [27]. Other studies have suggested that individuals may benefit from human-caused noise via a noise-induced predator shield [8,33] and the same could be true for other interactions. If this influence of altered soundscapes is widespread, we may discover unexpected preferences for noisy environments and reveal an additional driver of urban evolution.

One factor that may keep hosts from preferring noisy habitats is that mate attraction would be decreased via similar pathways that deter parasitoids. At least one study provides evidence that female field crickets (*Gryllus bimaculatus*) show impaired phonotaxis toward male song in noise [45]. Over evolutionary time, females of some host species or populations might adapt to noisy conditions by partially closing spiracles to change their sensitive frequency range [46,47] and develop neuronal detectors to distinguish conspecific songs in noise [48]. Such changes could improve male call detection under difficult listening conditions [49]. In one experiment, female attraction to male calls that spectrally overlapped with background noise was not hindered, suggesting that conspecifics may be able to adapt, behaviourally or via heritable change, to noisy conditions [50]. Whether these changes could result in population differentiation across both natural and anthropogenic gradients is an intriguing prospect, and remains to be investigated. Selection may act on signallers as well as receivers. For example, stratification of insect calls in noisy rainforests indicates that crickets have the ability to avoid biotic masking (e.g. [51,52]). Therefore, crickets and other insects may stratify calls to avoid anthropogenic masking as well, using a number of strategies such as spatial release, amplitude adjustments, temporal adjustments [53], or even switching to non-auditory cues, such as olfaction [54]. Whether or not signallers, receivers and parasitoids such as *O. ochracea* keep up with the selection pressure of increasing anthropogenic noise is likely to be a fruitful line of inquiry in animal behaviour research.

# 5. Conclusion

Overall, our study suggests that anthropogenic and natural noise have the potential to influence host–parasitoid interactions. Although the mechanism of why *O. ochracea* phonotaxis to hosts is impaired

by noise is undetermined, our results indicate that there may be a benefit to hosts living in noisy areas. Future studies investigating whether noise masks temporal or frequency information in cricket signals, disorients flies or simply triggers avoidance behaviour will be key to understanding how noise affects the antagonistic coevolution of parasitoid–host interactions.

Data accessibility. All data used in the analyses are included in the electronic supplementary material and are available at rs.figshare.com.

Authors' contributions. J.N.P. and S.K.R. wrote the manuscript with feedback from all other authors. C.D.F. designed the study. C.M.G., S.K.R. and M.-L.T.L. conducted the fieldwork. J.N.P., S.K.R. and C.D.F. analysed the data.

Competing interests. We have no competing interests.

Funding. We are grateful to National Science Foundation grant nos. CNH-1414171 and DEB-1556192 to C.D.F., DBI-1812280 to J.N.P., and a STEM Teacher and Research Fellowship to C.M.G.

Acknowledgements. We thank Carl LaRiccia and Wren Thompson for field assistance and Brian Leavell and three anonymous reviewers for helpful comments on earlier drafts of this manuscript.

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
