## [Reviewer comments · Royal Society Open Science]

Review History

RSOS-190867.R0 (Original submission)

Review form: Reviewer 1

Is the manuscript scientifically sound in its present form?

Yes

Are the interpretations and conclusions justified by the results?

No

Is the language acceptable?

Yes

Is it clear how to access all supporting data?

Yes

Do you have any ethical concerns with this paper?

No

Have you any concerns about statistical analyses in this paper?

Yes

Recommendation?

Accept with minor revision (please list in comments)

Comments to the Author(s)

Here, the authors investigate the impacts of various levels of traffic (treatment) and ocean surface (positive control) noise on the ability of *O. ochracea* to locate experimental playbacks of *G. lineaticeps*. The authors found that fewer *O. ochracea* were caught at traps exposed to greater levels of playback noise. The authors interpret this reduction in *O. ochracea* abundance at noisier traps as the potential for noise to interfere with host-parasite interactions. While I think this question is interesting and valid, the authors make too strong of claims with regards to the inferences that can be made based on their results. Specifically, the authors do not report the effects of noise on the distribution of *G. lineaticeps*. Therefore, the reader isn't able to evaluate the degree to which noise would in fact alter host-parasite interactions in this habitat. If hosts and parasites both avoid noise-exposed habitats, noise may in fact increase parasitism rates due to reduced habitat availability for both parasite and host rather than hosts benefitting from noise through reduced parasitism.

Major comments:

- Might field crickets, in addition to *O. ochracea*, avoid noisy areas? It seems like it's important to know this information (whether field crickets avoid noisy areas or not), in order to put these results in the larger context. For example, if both field crickets and their parasites avoid noisier areas - couldn't noise functionally increase parasitism rates because the amount of suitable habitat would be reduced by noise (and so the density of the parasite and host would increase in suitable habitats?) To fully address this concern, the authors would need to do a follow up study, sampling both cricket and parasite abundance in areas experimentally exposed to traffic and ocean surface noise. If this follow up study is not possible - the authors should, at the very least, acknowledge this possibility in their discussion (around line 235, where the authors suggest future studies investigate cricket abundance in noisier areas), and perhaps their introduction (background info on noise impacts on cricket abundance/distribution).

- The final paragraph of the discussion (Line 249 - 264) is too speculative and tangential in my opinion. It should be removed or reworked.

- Because you only have 2 years, 'year' should be included in the model as a fixed, not random, effect. Generally, you need more than 5 levels of a variable to avoid imprecise variance estimates (see Harrison et al 2018).

Harrison, X. A., Donaldson, L., Correa-Cano, M. E., Evans, J., Fisher, D. N., Goodwin, C. E., ... & Inger, R. (2018). A brief introduction to mixed effects modelling and multi-model inference in ecology. *PeerJ*, 6, e4794.

Minor comments:

Line 36: Clarify what you mean by 'increases in noise' - amplitude?

Lines 53 - 54: The phrase '...and fewer have focused on other interspecific interactions [8]...' is vague enough that it's not useful to the reader. Are you eluding to host/parasite interactions here? Either add more detail or remove.

Lines 62-64: This sentence is a restatement of the conclusion sentence in the previous paragraph (Lines 52-54). Either change one of these sentences, or perhaps combine the paragraphs?

Line 117 - Were there any differences in the frequency ranges between male songs? Also, it'd be useful to provide the reader with information on the general frequency range in which crickets sing.

Line 135: Please state the noise gradient here. From figure 5, it looks like levels varied between 40 - 70 dBA?

Line 239 – 240: I'm confused as to why the authors say that the impact of anthropogenic noise on host-parasite interactions is novel, given that they cite 2 studies in the previous paragraph that have already published on this topic. Is it that the previous studies just looked at noise v. control, rather than a gradient of noise? Make this distinction clearer or remove this sentence.

Review form: Reviewer 2

Is the manuscript scientifically sound in its present form?

Yes

Are the interpretations and conclusions justified by the results?

Yes

Is the language acceptable?

Yes

Is it clear how to access all supporting data?

Yes

Do you have any ethical concerns with this paper?

No

Have you any concerns about statistical analyses in this paper?

Yes

Recommendation?

Accept with minor revision (please list in comments)

Comments to the Author(s)

This study aims to examine the effects of noise on the phonotactic behaviour of a parasitoid fly, *Ormia ochracea*, which localise its cricket host on the basis of its calling song. The Authors performed field playbacks of two types of noise (traffic noise and ocean surf noise) and cricket calling song samples was broadcasted from speakers placed at a range of distances from the experimental noise source. Their results show that the number of flies approached the cricket song producing speakers decreased with increasing background noise level.

It is a well written manuscript presenting interesting results obtained using mainly appropriate methods and based on a sufficient sample size. However there are some points of the study design and methods, which arise questions.

1. Crickets may also react to noise. If they manage to alter their songs to avoid or to decrease the masking effect of noise, flies may find them more efficiently than predicted by this study. Or, on the contrary, if crickets cease to sing at some levels of noise, flies will be even less capable of finding their host than forecasted by the results of this study. Of course, this problem can not be solved at this stage of this research, but a short paragraph about the possible effect of cricket song modification in reaction to noise could be added to the discussion.

2. Cricket songs were played back using two different types of speaker (STORMp3 or Satechi SD Mini portable speakers), noise samples were emitted using other two different types of speakers (TIC GS5 and Block Rocker). The Authors present two figures showing that the frequency spectra

of the emitted sound samples are very similar to the frequency spectra of the original cricket song recordings and to that of the original noise samples. Unfortunately it is a much more complex question whether or not different speakers differ in attracting flies or in reproducing the sound field around a noise source: for example the sound radiation pattern of the different speakers may well be an important feature, their impulse response characteristics may also modify the results, and even more characteristics may have some importance. Therefore I think it would be more convincing and more straightforward to include the applied speaker type in the glm models as a fixed effect and test if the inclusion of speaker type as a factor has a significant effect.

3. Cricket call rate is an important song feature for the flies to choose amongst male crickets. The Authors examined if adding call rate as a fixed effect improved model fit. They found it did not. It would be interesting to test also the interaction between noise level and the effect of call rate, since flies may be less capable of hearing the difference between the calling song samples when the background noise level is high.

4. It would be important to include the obtained model parameter estimates (intercept, regression slope, the effects of fixed effect of factor levels) and their statistics in the results, because readers how are not familiar with using R may have difficulties to access those results.

5. Did the authors examine the possibility that the number of fly specimens caught by the traps follow a Poisson rather than a normal distribution?

6. How many noise samples were recorded? What was the duration of the sound file composed from the noise samples for the noise playback? Why the authors decided to normalise noise samples to the same peak amplitude? That way the amplification of recordings depended on a relatively short acoustic event. A measure of the average noise level (e.g RMS amplitude, or the Leq sound level) could have been a better choice to adjust the amplitude of the samples to the same amplitude.

7. It would be enough to present the glm results of the whole data set to show that noise affected negatively the number of flies caught and there was no difference in the effects of traffic noise versus ocean surf noise. Why did the authors decide to present also the results of an analysis based on the data subset containing only the traffic noise playback?

8. At lines 221-223. The Authors write: "Furthermore, *O. ochracea* can detect lower frequency sounds at higher amplitude thresholds [37], thus low frequency energy may stimulate tympanal membranes and interfere with the fly's ability to detect higher frequency informative signals." The first statement is supported by the results of the referred paper, but the second statement do not follow from that. Could the Authors, please, give a reference on which the second statement is based?

Since I am not a native English speaker I do not try to make suggestions how to improve the text in details.

Review form: Reviewer 3

Is the manuscript scientifically sound in its present form?

Yes

Are the interpretations and conclusions justified by the results?

Yes

Is the language acceptable?

Yes

Is it clear how to access all supporting data?

Yes

Do you have any ethical concerns with this paper?

No

Have you any concerns about statistical analyses in this paper?

No

Recommendation?

Accept with minor revision (please list in comments)

Comments to the Author(s)

I was delighted to read this manuscript, as expanding our understanding of the impacts of anthropogenic noise on animals to interspecific interactions is an important step. The cricket-fly interaction is a perfect opportunity to assess the impacts of noise on parasitism and I really liked the addition of ocean sounds to the experimental design. I do think that the introduction presents the work as potentially more novel than it is – it would be nice to acknowledge the work that has already been done on influences of traffic noise on host/parasite(parasitoid) interactions more completely in the introduction (as is done in the discussion). Aside from that I have only minor comments, which I've outlined below.

Line 83: reword sentence – reduced choosiness does not necessarily follow from females being parasitized making the word “thus” less appropriate

Line 84: This sentence seems incomplete – please link the phrase “for such a small animal” to the rest of the sentence. I assume the authors are referring to the unique hearing mechanism the flies use, given their size.

Line 94: please describe the potential for anthropogenic noise to mask cricket calls in the introduction rather than just the figure. This is an important point. The reader needs to fully understand why masking is likely.

Line 194: This is a perfect opportunity to include an effect size in the text – can you provide some information about by what % traffic/ocean waves reduce attraction of parasitoid flies, given a particular increase in noise? I think this is particularly important given than the pattern in the figure doesn't look particularly dramatic (not to say the finding isn't important – I'm convinced).

Decision letter (RSOS-190867.R0)

08-Aug-2019

Dear Dr Phillips

On behalf of the Editors, I am pleased to inform you that your Manuscript RSOS-190867 entitled "Background acoustics disrupt host-parasitoid interactions" has been accepted for publication in

Royal Society Open Science subject to minor revision in accordance with the referee suggestions. Please find the referees' comments at the end of this email.

The reviewers and handling editors have recommended publication, but also suggest some minor revisions to your manuscript. Therefore, I invite you to respond to the comments and revise your manuscript.

- Ethics statement

- Data accessibility

If you wish to submit your supporting data or code to Dryad (<http://datadryad.org/>), or modify your current submission to dryad, please use the following link:
<http://datadryad.org/submit?journalID=RSOS&manu=RSOS-190867>

- Competing interests

- Authors' contributions

- Acknowledgements

- Funding statement

Because the schedule for publication is very tight, it is a condition of publication that you submit the revised version of your manuscript before 17-Aug-2019. Please note that the revision deadline will expire at 00.00am on this date. If you do not think you will be able to meet this date please let me know immediately.

Supplementary files will be published alongside the paper on the journal website and posted on the online figshare repository (<https://rs.figshare.com/>). The heading and legend provided for each supplementary file during the submission process will be used to create the figshare page,

so please ensure these are accurate and informative so that your files can be found in searches. Files on figshare will be made available approximately one week before the accompanying article so that the supplementary material can be attributed a unique DOI.

Kind regards,

on behalf of Professor Kevin Padian (Subject Editor)
openscience@royalsociety.org

Editor Comments to Author:

Thanks for your submission. The reviewers were overall happy with your efforts and we urge you to consider their suggestions in finalizing your manuscript.

Reviewer comments to Author:

Reviewer: 1
Comments to the Author(s)

Here, the authors investigate the impacts of various levels of traffic (treatment) and ocean surface (positive control) noise on the ability of *O. ochracea* to locate experimental playbacks of *G. lineaticeps*. The authors found that fewer *O. ochracea* were caught at traps exposed to greater levels of playback noise. The authors interpret this reduction in *O. ochracea* abundance at noisier traps as the potential for noise to interfere with host-parasite interactions. While I think this question is interesting and valid, the authors make too strong of claims with regards to the inferences that can be made based on their results. Specifically, the authors do not report the effects of noise on the distribution of *G. lineaticeps*. Therefore, the reader isn't able to evaluate the degree to which noise would in fact alter host-parasite interactions in this habitat. If hosts and parasites both avoid noise-exposed habitats, noise may in fact increase parasitism rates due to reduced habitat availability for both parasite and host rather than hosts benefitting from noise through reduced parasitism.

Major comments:

- Might field crickets, in addition to *O. ocracea*, avoid noisy areas? It seems like it's important to know this information (whether field crickets avoid noisy areas or not), in order to put these results in the larger context. For example, if both field crickets and their parasites avoid noisier areas – couldn't noise functionally increase parasitism rates because the amount of suitable habitat would be reduced by noise (and so the density of the parasite and host would increase in suitable habitats?) To fully address this concern, the authors would need to do a follow up study, sampling both cricket and parasite abundance in areas experimentally exposed to traffic and ocean surface noise. If this follow up study is not possible – the authors should, at the very least, acknowledge this possibility in their discussion (around line 235, where the authors suggest future studies investigate cricket abundance in noisier areas), and perhaps their introduction (background info on noise impacts on cricket abundance/distribution).
- The final paragraph of the discussion (Line 249 – 264) is too speculative and tangential in my opinion. It should be removed or reworked.
- Because you only have 2 years, 'year' should be included in the model as a fixed, not random, effect. Generally, you need more than 5 levels of a variable to avoid imprecise variance estimates (see Harrison et al 2018).

Harrison, X. A., Donaldson, L., Correa-Cano, M. E., Evans, J., Fisher, D. N., Goodwin, C. E., ... & Inger, R. (2018). A brief introduction to mixed effects modelling and multi-model inference in ecology. *PeerJ*, 6, e4794.

Minor comments:

Line 36: Clarify what you mean by 'increases in noise' – amplitude?

Lines 53 – 54: The phrase '...and fewer have focused on other interspecific interactions [8]...' is vague enough that it's not useful to the reader. Are you eluding to host/parasite interactions here? Either add more detail or remove.

Lines 62-64: This sentence is a restatement of the conclusion sentence in the previous paragraph (Lines 52-54). Either change one of these sentences, or perhaps combine the paragraphs?

Line 117 – Were there any differences in the frequency ranges between male songs? Also, it'd be useful to provide the reader with information on the general frequency range in which crickets sing.

Line 135: Please state the noise gradient here. From figure 5, it looks like levels varied between 40 – 70 dBA?

Line 239 – 240: I'm confused as to why the authors say that the impact of anthropogenic noise on host-parasite interactions is novel, given that they cite 2 studies in the previous paragraph that have already published on this topic. Is it that the previous studies just looked at noise v. control, rather than a gradient of noise? Make this distinction clearer or remove this sentence.

Reviewer: 2

Comments to the Author(s)

This study aims to examine the effects of noise on the phonotactic behaviour of a parasitoid fly, *Ormia ochracea*, which localise its cricket host on the basis of its calling song. The Authors performed field playbacks of two types of noise (traffic noise and ocean surf noise) and cricket calling song samples was broadcasted from speakers placed at a range of distances from the experimental noise source. Their results show that the number of flies approached the cricket song producing speakers decreased with increasing background noise level.

It is a well written manuscript presenting interesting results obtained using mainly appropriate

methods and based on a sufficient sample size. However there are some points of the study design and methods, which arise questions.

1. Crickets may also react to noise. If they manage to alter their songs to avoid or to decrease the masking effect of noise, flies may find them more efficiently than predicted by this study. Or, on the contrary, if crickets cease to sing at some levels of noise, flies will be even less capable of finding their host than forecasted by the results of this study. Of course, this problem can not be solved at this stage of this research, but a short paragraph about the possible effect of cricket song modification in reaction to noise could be added to the discussion.

2. Cricket songs were played back using two different types of speaker (STORMp3 or Satechi SD Mini portable speakers), noise samples were emitted using other two different types of speakers (TIC GS5 and Block Rocker). The Authors present two figures showing that the frequency spectra of the emitted sound samples are very similar to the frequency spectra of the original cricket song recordings and to that of the original noise samples. Unfortunately it is a much more complex question whether or not different speakers differ in attracting flies or in reproducing the sound field around a noise source: for example the sound radiation pattern of the different speakers may well be an important feature, their impulse response characteristics may also modify the results, and even more characteristics may have some importance. Therefore I think it would be more convincing and more straightforward to include the applied speaker type in the glm models as a fixed effect and test if the inclusion of speaker type as a factor has a significant effect.

3. Cricket call rate is an important song feature for the flies to choose amongst male crickets. The Authors examined if adding call rate as a fixed effect improved model fit. They found it did not. It would be interesting to test also the interaction between noise level and the effect of call rate, since flies may be less capable of hearing the difference between the calling song samples when the background noise level is high.

4. It would be important to include the obtained model parameter estimates (intercept, regression slope, the effects of fixed effect of factor levels) and their statistics in the results, because readers how are not familiar with using R may have difficulties to access those results.

5. Did the authors examine the possibility that the number of fly specimens caught by the traps follow a Poisson rather than a normal distribution?

6. How many noise samples were recorded? What was the duration of the sound file composed from the noise samples for the noise playback? Why the authors decided to normalise noise samples to the same peak amplitude? That way the amplification of recordings depended on a relatively short acoustic event. A measure of the average noise level (e.g RMS amplitude, or the Leq sound level) could have been a better choice to adjust the amplitude of the samples to the same amplitude.

7. It would be enough to present the glm results of the whole data set to show that noise affected negatively the number of flies caught and there was no difference in the effects of traffic noise versus ocean surf noise. Why did the authors decide to present also the results of an analysis based on the data subset containing only the traffic noise playback?

8. At lines 221-223. The Authors write: "Furthermore, *O. ochracea* can detect lower frequency sounds at higher amplitude thresholds [37], thus low frequency energy may stimulate tympanal membranes and interfere with the fly's ability to detect higher frequency informative signals." The first statement is supported by the results of the referred paper, but the second statement do not follow from that. Could the Authors, please, give a reference on which the second statement is based?

Since I am not a native English speaker I do not try to make suggestions how to improve the text in details.

Reviewer: 3

Comments to the Author(s)

I was delighted to read this manuscript, as expanding our understanding of the impacts of anthropogenic noise on animals to interspecific interactions is an important step. The cricket-fly interaction is a perfect opportunity to assess the impacts of noise on parasitism and I really liked the addition of ocean sounds to the experimental design. I do think that the introduction presents the work as potentially more novel than it is – it would be nice to acknowledge the work that has already been done on influences of traffic noise on host/ parasite(parasitoid) interactions more completely in the introduction (as is done in the discussion). Aside from that I have only minor comments, which I've outlined below.

Line 83: reword sentence – reduced choosiness does not necessarily follow from females being parasitized making the word “thus” less appropriate

Line 84: This sentence seems incomplete – please link the phrase “for such a small animal” to the rest of the sentence. I assume the authors are referring to the unique hearing mechanism the flies use, given their size.

Line 94: please describe the potential for anthropogenic noise to mask cricket calls in the introduction rather than just the figure. This is an important point. The reader needs to fully understand why masking is likely.

Line 194: This is a perfect opportunity to include an effect size in the text – can you provide some information about by what % traffic/ocean waves reduce attraction of parasitoid flies, given a particular increase in noise? I think this is particularly important given than the pattern in the figure doesn't look particularly dramatic (not to say the finding isn't important – I'm convinced).

Author's Response to Decision Letter for (RSOS-190867.R0)

See Appendix A.

Decision letter (RSOS-190867.R1)

27-Aug-2019

Dear Dr Phillips,

I am pleased to inform you that your manuscript entitled "Background noise disrupts host-parasitoid interactions" is now accepted for publication in Royal Society Open Science.

Kind regards,

on behalf of Professor Kevin Padian (Subject Editor)
openscience@royalsociety.org

Appendix A

Dear Dr Phillips

On behalf of the Editors, I am pleased to inform you that your Manuscript RSOS-190867 entitled "Background acoustics disrupt host-parasitoid interactions" has been accepted for publication in Royal Society Open Science subject to minor revision in accordance with the referee suggestions. Please find the referees' comments at the end of this email.

The reviewers and handling editors have recommended publication, but also suggest some minor revisions to your manuscript. Therefore, I invite you to respond to the comments and revise your manuscript.

- Ethics statement

- Data accessibility

<http://datadryad.org/submit?journalID=RSOS&manu=RSOS-190867>

- Competing interests

- Authors' contributions

- Acknowledgements

- Funding statement

Because the schedule for publication is very tight, it is a condition of publication that you submit the revised version of your manuscript before 17-Aug-2019. Please note that the revision deadline will expire at 00.00am on this date. If you do not think you will be able to meet this date please let me know immediately.

- 1) A text file of the manuscript (tex, txt, rtf, docx or doc), references, tables (including captions) and figure captions. Do not upload a PDF as your "Main Document";

- 2) A separate electronic file of each figure (EPS or print-quality PDF preferred (either format should be produced directly from original creation package), or original software format);
- 3) Included a 100 word media summary of your paper when requested at submission. Please ensure you have entered correct contact details (email, institution and telephone) in your user account;
- 4) Included the raw data to support the claims made in your paper. You can either include your data as electronic supplementary material or upload to a repository and include the relevant doi within your manuscript. Make sure it is clear in your data accessibility statement how the data can be accessed;
- 5) All supplementary materials accompanying an accepted article will be treated as in their final form. Note that the Royal Society will neither edit nor typeset supplementary material and it will be hosted as provided. Please ensure that the supplementary material includes the paper details where possible (authors, article title, journal name).

Kind regards,

on behalf of Professor Kevin Padian (Subject Editor)
openscience@royalsociety.org

Editor Comments to Author:

Thanks for your submission. The reviewers were overall happy with your efforts and we urge you to consider their suggestions in finalizing your manuscript.

Reviewer comments to Author:

Reviewer: 1

Comments to the Author(s)

Here, the authors investigate the impacts of various levels of traffic (treatment) and ocean surface (positive control) noise on the ability of *O. ochracea* to locate experimental playbacks of *G. lineaticeps*. The authors found that fewer *O. ochracea* were caught at traps exposed to greater levels of playback noise. The authors interpret this reduction in *O. ochracea* abundance at noisier traps as the potential for noise to interfere with host-parasite interactions. While I think this question is interesting and valid, the authors make too strong of claims with regards to the inferences that can be made based on their results. Specifically, the authors do not report the effects of noise on the distribution of *G. lineaticeps*. Therefore, the reader isn't able to evaluate the degree to which noise would in fact alter host-parasite interactions in this habitat. If hosts and parasites both avoid noise-exposed habitats, noise may in fact increase parasitism rates due to reduced habitat availability for both parasite and host rather than hosts benefitting from noise through reduced parasitism.

RESPONSE: Thank you for your comments. We have addressed your concerns below.

Major comments:

- Might field crickets, in addition to *O. ochracea*, avoid noisy areas? It seems like it's important to know this information (whether field crickets avoid noisy areas or not), in order to put these results in the larger context. For example, if both field crickets and their parasites avoid noisier areas – couldn't noise functionally increase parasitism rates because the amount of suitable habitat would be reduced by noise (and so the density of the parasite and host would increase in suitable habitats?) To fully address this concern, the authors would need to do a follow up study, sampling both cricket and parasite abundance in areas experimentally exposed to traffic and ocean surface noise. If this follow up study is not possible – the authors should, at the very least, acknowledge this possibility in their discussion (around line 235, where the authors suggest future studies investigate cricket abundance in noisier areas), and perhaps their introduction (background info on noise impacts on cricket abundance/distribution).

RESPONSE: We now address this concern in this paragraph, adding (Line 265) “Additional work should investigate whether crickets are more abundant in noise and have fewer parasitoid infections to support the hypothesis that hosts may benefit from noise.

- The final paragraph of the discussion (Line 249 – 264) is too speculative and tangential in my opinion. It should be removed or reworked.

RESPONSE: This paragraph addresses potential outcomes of noise disrupting sexual selection of hosts, and provides evidence with current literature in this area. We have reworked this paragraph to reflect supporting literature for the idea that noise can act as a selective pressure on signals and signalers, and to emphasize that much remains to be done in future studies. It now reads (LINE 278):

“One factor that may keep hosts from preferring noisy habitats is that mate attraction would be decreased via similar pathways that deter parasitoids. At least one study provides evidence that female field crickets (*Gryllus bimaculatus*) show impaired phonotaxis toward male song in noise [46]. Over evolutionary time, females of some host species or populations might adapt to noisy conditions by partially closing spiracles to change their sensitive frequency range [47,48] and develop neuronal detectors to distinguish conspecific songs in noise [49]. Such changes could improve male call detection under difficult listening conditions [50]. In one experiment, female attraction to male calls that spectrally overlapped with background noise was not hindered, suggesting that conspecifics may be able to adapt, behaviorally or via heritable change, to noisy conditions [51]. Whether these changes could result in population differentiation across both natural and anthropogenic gradients is an intriguing prospect, and remains to be investigated. Selection may act on signalers as well as receivers. For example, stratification of insect calls in noisy rainforests indicates that crickets have the ability to avoid biotic masking [e.g. 40,41]. Therefore, crickets and other insects may stratify calls to avoid anthropogenic masking as well, using a number of strategies such as spatial release, amplitude adjustments, temporal adjustments [54], or even switching to non-auditory cues, such as olfaction [55]. Whether or not signalers, receivers, and parasitoids such as *O. ochracea* keep up with the selection pressure of increasing anthropogenic noise is likely to be a fruitful line of inquiry in animal behavior research.”

- Because you only have 2 years, ‘year’ should be included in the model as a fixed, not random, effect. Generally, you need more than 5 levels of a variable to avoid imprecise variance estimates (see Harrison et al 2018).

Harrison, X. A., Donaldson, L., Correa-Cano, M. E., Evans, J., Fisher, D. N., Goodwin, C. E., ... & Inger, R. (2018). A brief introduction to mixed effects modelling and multi-model inference in ecology. PeerJ, 6, e4794.

RESPONSE: We now include Year, which also correlates with speaker type, as a fixed effect in our models, and have reanalyzed all models without year as a random effect. We also now state why different speakers were used between years (line 151) and emphasize that any potential differences between years could be latent environmental differences or possible differences in background noise playback between the two systems. Year/Speaker type does not improve model fit; we now report this in the results section (line 226): “Additionally, inclusion of speaker type as a fixed affect did not significantly improve model fit for traffic only ($\chi^2 = 3.18$, $p = 0.07$, $df = 1$) or for traffic and ocean noise trials combined ($\chi^2 = 2.68$, $p = 0.10$, $df = 1$).”

Minor comments:

Line 36: Clarify what you mean by ‘increases in noise’ – amplitude?

RESPONSE: We now clarify by stating (Line 37) ‘increases in noise amplitude’

Lines 53 – 54: The phrase ‘...and fewer have focused on other interspecific interactions [8]...’ is vague enough that it’s not useful to the reader. Are you eluding to host/parasite interactions here? Either add more detail or remove.

RESPONSE: We have changed this sentence to ‘...(line 56)’ and fewer have focused on other interspecific interactions, such as mutualism or parasitism [8,13,14].

Lines 62-64: This sentence is a restatement of the conclusion sentence in the previous paragraph (Lines 52-54). Either change one of these sentences, or perhaps combine the paragraphs?

RESPONSE: We have combined these paragraphs since they both broadly introduce the topic and the gaps in the current literature (Line 44-64).

Line 117 – Were there any differences in the frequency ranges between male songs? Also, it’d be useful to provide the reader with information on the general frequency range in which crickets sing.

RESPONSE: There were no significant differences in stimulus frequency. We now add a sentence describing that songs were in the normal frequency range (line 132): “Therefore, the 20 randomly selected files used in our trials (see below) were varied somewhat in total length (mean \pm SD seconds = 28.82 ± 8.46 , range = 18.5—52.66), and fall within the normal range of cricket song frequency (3-8 kHz).”

Line 135: Please state the noise gradient here. From figure 5, it looks like levels varied between 40 – 70 dBA?

RESPONSE: We now state (Line 148): “We used either an ION Block Rocker (frequency response: 65Hz- 20kHz) or TIC GS5 speakers (frequency response: 55Hz- 20kHz), which played a unique traffic recording to establish a gradient in background noise levels, which ranged from 39.2 to 65.8 dB(A) *Leq*.”

Line 239 – 240: I’m confused as to why the authors say that the impact of anthropogenic noise on host-parasite interactions is novel, given that they cite 2 studies in the previous paragraph that have already published on this topic. Is it that the previous studies just looked at noise v. control, rather than a gradient of noise? Make this distinction clearer or remove this sentence.

RESPONSE: We have revised this sentence (line 269): “Our finding that risk of parasitoid infection may depend on acoustic conditions adds to a small body of work in animal behavior and evolutionary biology, and future research should parse out whether patterns of parasitoid infestation in the wild can be explained by variation in noise levels.”

Reviewer: 2

Comments to the Author(s)

This study aims to examine the effects of noise on the phonotactic behaviour of a parasitoid fly, *Ormia ochracea*, which localise its cricket host on the basis of its calling song. The Authors performed field playbacks of two types of noise (traffic noise and ocean surf noise) and cricket calling song samples was broadcasted from speakers placed at a range of distances from the experimental noise source. Their results show that the number of flies approached the cricket song producing speakers decreased with increasing background noise level.

It is a well written manuscript presenting interesting results obtained using mainly appropriate methods and based on a sufficient sample size. However there are some points of the study design and methods, which arise questions.

1. Crickets may also react to noise. If they manage to alter their songs to avoid or to decrease the masking effect of noise, flies may find them more efficiently than predicted by this study. Or, on the contrary, if crickets cease to sing at some levels of noise, flies will be even less capable of finding their host than forecasted by the results of this study. Of course, this problem can not be solved at this stage of this research, but a short paragraph about the possible effect of cricket song modification in reaction to noise could be added to the discussion.

RESPONSE: We have addressed this issue as indicated to reviewer one in line 265 and the last two paragraphs of the discussion.

2. Cricket songs were played back using two different types of speaker (STORMp3 or Satechi SD Mini portable speakers), noise samples were emitted using other two different types of speakers (TIC GS5 and Block Rocker). The Authors present two figures showing that the frequency spectra of the emitted sound samples are very similar to the frequency spectra of the original cricket song recordings and to that of the original noise samples. Unfortunately it is a much more complex question whether or not different speakers differ in attracting flies or in reproducing the sound field around a noise source: for example the sound radiation pattern of the different speakers may well be an important feature, their impulse response characteristics may also modify the results, and even more characteristics may have some importance. Therefore I think it would be more convincing and more straightforward to include the applied speaker type in the glm models as a fixed effect and test if the inclusion of speaker type as a factor has a significant effect.

RESPONSE: Thank you for your thorough review of the methodology. Upon review with our co-authors that conducted the trials, we found that only the Satechi SD mini portable speaker was used for the cricket song playback—the Stormp3 speakers were taken to the field for deployment if and when a Satechi speaker failed. This did not occur, therefore the Stormp3 speakers were never used for this experiment. Therefore, we have removed any reference to these speakers (line 164). We did use two speaker types for the noise playback. Because the Block Rockers batteries began to fail after the first season, we switched to the TIC GS5 speakers in 2017. We now include the speaker type used for the noise playback in the fixed effects models (line 194).

Speaker type does not significantly improve the model fit, but we now report the lesser models testing the effect of speaker type/year (Line 225-227):

“Additionally, inclusion of speaker type as a fixed effect did not significantly improve model fit for traffic only ($\chi^2 = 3.18$, $p = 0.07$, $df = 1$) or for traffic and ocean noise trials combined ($\chi^2 = 2.68$, $p = 0.10$, $df = 1$).”

3. Cricket call rate is an important song feature for the flies to choose amongst male crickets. The Authors examined if adding call rate as a fixed effect improved model fit. They found it did not. It would be interesting to test also the interaction between noise level and the effect of call rate, since flies may be less capable of hearing the difference between the calling song samples when the background noise level is high.

RESPONSE: We now include analyses of models that includes the interaction of call rate and noise level. Model fit does not improve. We now write (Line 228): “Finally, inclusion of cricket playback call rate as a fixed effect did not improve model performance (traffic only, $\chi^2 = 0.48$, $p = 0.49$, $df = 1$; traffic and ocean noise, $\chi^2 = 0.63$, $p = 0.43$, $df = 1$), nor did the interaction between call rate and noise improve model fit relative to a model with noise level only (traffic only, $\chi^2 = 1.262$, $p = 0.532$, $df = 2$; traffic and ocean noise, $\chi^2 = 1.174$, $p = 0.556$, $df = 2$). Similarly, call rate without playback stimuli as a random effect did not improve model fit over the model with just noise as a fixed effect (traffic only, $\chi^2 = 0.920$, $p = 0.337$, $df = 1$; traffic and ocean noise, $\chi^2 = 1.061$, $p = 0.303$, $df = 1$).”

4. It would be important to include the obtained model parameter estimates (intercept, regression slope, the effects of fixed effect of factor levels) and their statistics in the results, because readers how are not familiar with using R may have difficulties to access those results.

RESPONSE: We now include a table of the best model fit with these values for the models of best fit—Table 1 for traffic noise only and Table 2 for all noise treatments.

Table 1. Final model parameters for traffic noise trials

Random effects	Variance	SD			
Stimulus	0.069	0.263			
Trial	0.209	0.457			
Residual	0.357	0.597			
Fixed Effects	Estimate	SE	df	t-value	p
(Intercept)	3.496	0.706	59.970	4.954	<0.001
Noise Level (dBA)	-0.034	0.014	58.080	-2.517	0.014

Table 2. Final model parameters for both ocean and traffic noise trials

Random effects	Variance	SD			
Stimulus	0.066	0.258			
Trial	0.173	0.416			
Residual	0.328	0.572			
Fixed Effects	Estimate	SE	df	t-value	p
(Intercept)	3.531	0.629	75.787	5.615	<0.001
Noise Level (dBA)	-0.035	0.012	76.934	-2.940	0.004

5. Did the authors examine the possibility that the number of fly specimens caught by the traps follow a Poisson rather than a normal distribution?

RESPONSE: In a preliminary analysis we modeled the number of flies caught with a generalized linear mixed effect model and Poisson error, but found overdispersion to be too high for model assumptions. We then found that log transformation of the response variable, adjusted by +1, meet assumptions of normality for use in the linear mixed effect models we used in this paper. This is now stated in the methods section (Line 181): “In preliminary analyses we had explored generalized linear mixed effect models with Poisson error, given that number of flies caught reflects counts. However, models consistently suffered from overdispersion. Therefore, we used LMMs and for each model we log transformed number of *O. ochracea* following a quantitative adjustment of adding one to all observations.”

6. How many noise samples were recorded? What was the duration of the sound file composed from the noise samples for the noise playback? Why the authors decided to normalise noise samples to the same peak amplitude? That way the amplification of recordings depended on a relatively short acoustic event. A measure of the average noise level (e.g RMS amplitude, or the Leq sound level) could have been a better choice to adjust the amplitude of the samples to the same amplitude.

RESPONSE: We used seven types of noise files, 6 traffic noise and one surf noise file. Traffic noise recordings averaged 200 s, whereas the single surf noise playback file was 1-hr in length and was constructed as a composite average of multiple recordings of surf noise recorded throughout the Central California Coast. The six traffic noise files were standardized to the same peak amplitude to minimize the potential influence of infrequent but very loud sound events causing behavioral changes (reviewed in Francis & Barber 2013). The reviewer is correct that standardization based on RMS or Leq would have allowed better standardization to the same amplitude, but this was not the goal. Instead, we wanted the noise playbacks to vary so that sound level at each trap could be used as a continuous predictor variable in our models.

The paragraph describing noise files now reads (line 139): “To simulate anthropogenic noise, we recorded six traffic noise samples throughout San Luis Obispo County using Roland R05 recorders at a distance of 10 m from roadways. Recordings averaged 200 ± 16 (SE) seconds and

were standardized to the same peak amplitude in Raven Pro 1.5 [33]. To each recording we added a 5 second fade in and fade out to the beginning and end of each to eliminate rapid onset and offset as the recording playbacks looped continuously. To explore the possible difference between traffic noise and natural sources of background noise, we used ocean surf noise in a subset of experimental trials. To simulate ocean surf noise, we recorded ocean surf noise at four locations throughout the Central Coast of California at 10m from the edge of coastal bluffs using identical methods to those describe for traffic noise. To provide a stimulus typical of coastal-scrub environments on the Central Coast of California, we combined the four recordings into a single 60-min playback file and crossfaded the original four recordings with 5 second fade in and fade out so that the transition from one recording to the next was seamless.”

7. It would be enough to present the glm results of the whole data set to show that noise affected negatively the number of flies caught and there was no difference in the effects of traffic noise versus ocean surf noise. Why did the authors decide to present also the results of an analysis based on the data subset containing only the traffic noise playback?

RESPONSE: We believe it is important to present both the combined ocean noise and traffic noise and traffic noise alone to show that noise type, whether manmade or natural, can both affect host-parasite interactions. Much attention has been paid effects of anthropogenic noise and by sequentially presenting these results we can, in our opinion, more convincingly demonstrate that natural sounds can have the same effect.

8. At lines 221-223. The Authors write: "Furthermore, *O. ochracea* can detect lower frequency sounds at higher amplitude thresholds [37], thus low frequency energy may stimulate tympanal membranes and interfere with the fly's ability to detect higher frequency informative signals." The first statement is supported by the results of the referred paper, but the second statement do not follow from that. Could the Authors, please, give a reference on which the second statment is based?

RESPONSE: We have changed this section to better reflect the literature and reasons why low frequency noise may still interfere with detection of high frequency signals (line 251): “Although most traffic and ocean noises are low frequency, substantial energy extends into *O. ochracea*'s best sensitivity and the peak frequency of variable field cricket calls (Fig. 1), which could decrease signal-to-noise ratios and render signal detection and localization difficult for the parasitoids via masking [31]. Low frequency noise may also act as a distraction from the target cricket calls [41,42], as evidenced by crabs that change their risk assessment behavior in noise to predators [43]. Additionally, even if low frequency noise does not directly mask cricket calls, it may impair the parasitoid's ability to detect variation in temporal patterns of the calls, which is demonstrated to be an important feature for *O. ochracea* attraction [21,36,44].”

Since I am not a native English speaker I do not try to make suggestions how to improve the text in details.

Reviewer: 3

Comments to the Author(s)

I was delighted to read this manuscript, as expanding our understanding of the impacts of anthropogenic noise on animals to interspecific interactions is an important step. The cricket-fly interaction is a perfect opportunity to assess the impacts of noise on parasitism and I really liked the addition of ocean sounds to the experimental design. I do think that the introduction presents the work as potentially more novel than it is – it would be nice to acknowledge the work that has already been done on influences of traffic noise on host/parasite(parasitoid) interactions more completely in the introduction (as is done in the discussion). Aside from that I have only minor comments, which I've outlined below.

RESPONSE: Thank you for your review. We now acknowledge the small body of literature that investigates noise on host/parasite interactions in Line 94 in the introduction: “For example, the call of the túngara frog is relatively low (1-3 kHz), and frog-biting midges (*Corethrella* sp.) that orient toward túngara calls are less abundant in noise [15].”

Line 83: reword sentence – reduced choosiness does not necessarily follow from females being parasitized making the word “thus” less appropriate

RESPONSE: We have changed this sentence to (line 83): “However, female crickets can also be parasitized, which may lower their choosiness for faster calls [26].”

Line 84: This sentence seems incomplete – please link the phrase “for such a small animal” to the rest of the sentence. I assume the authors are referring to the unique hearing mechanism the flies use, given their size.

RESPONSE: We have changed this sentence to (line 84) “A great deal of research has focused on how these flies hear and orient toward the cricket song. Given their small size, they are surprisingly able to localize a host with two degree accuracy by amplifying sound differences, similar to the hearing system of humans [29,30].”

Line 94: please describe the potential for anthropogenic noise to mask cricket calls in the introduction rather than just the figure. This is an important point. The reader needs to fully understand why masking is likely.

RESPONSE: We have added a paragraph emphasizing this point in the text (starting line 90): “Background noise from both natural and anthropogenic sources consists of a wide range of frequencies that overlap with animal vocalizations, which can mask those signals from being detected by receivers [31]. Typically, ocean noise and traffic noise are louder at low frequencies with decreasing energy at higher frequencies (Fig. 1). Therefore, most studies focus on how low frequency noise overlaps low frequency signals. For example, the call of the túngara frog is relatively low (1-3 kHz), and frog-biting midges (*Corethrella* sp.) that orient toward túngara calls are less abundant in noise [15]. Similarly, the relatively high acoustic energy in traffic and ocean surf noise at 5 kHz overlaps the peak frequency of cricket calls (Fig. 1). Given this

overlap, these sources of background noise are likely to at least partially mask the signal for receivers (Fig. 1).”

Line 194: This is a perfect opportunity to include an effect size in the text – can you provide some information about by what % traffic/ocean waves reduce attraction of parasitoid flies, given a particular increase in noise? I think this is particularly important given that the pattern in the figure doesn't look particularly dramatic (not to say the finding isn't important – I'm convinced).

RESPONSE: We now state at this line 217: “...such that the estimated number of flies caught at the loudest sound levels (approx. 65 dB) were more than 50% lower than the estimated numbers caught in most quiet conditions.”